# Video Stream Recognition Using Bitstream Shape for Mobile Network QoE

**DOI:** 10.3390/s23052548

**Published:** 2023-02-24

**Authors:** Darius Chmieliauskas, Šarūnas Paulikas

**Affiliations:** Department of Computer Science and Communications Technologies, Vilnius Gediminas Technical University, 10223 Vilnius, Lithuania

**Keywords:** mobile network traffic, encrypted video stream recognition, time series classification, QoS/QoE LTE 5G management

## Abstract

Video streaming service delivery is a challenging task for mobile network operators. Knowing which services clients are using could help ensure a specific quality of service and manage the users’ experience. Additionally, mobile network operators could apply throttle, traffic prioritization, or differentiated pricing. However, due to the growth of encrypted Internet traffic, it has become difficult for network operators to recognize the type of service used by their clients. In this article, we propose and evaluate a method for recognizing video streams solely based on the shape of the bitstream on a cellular network communication channel. To classify bitstreams, we used a convolutional neural network that was trained on a dataset of download and upload bitstreams collected by the authors. We demonstrate that our proposed method achieves an accuracy of over 90% in recognizing video streams from real-world mobile network traffic data.

## 1. Introduction

Mobile data traffic is increasing at a steady rate of 42% annually, with video traffic making up 69% of all mobile data traffic [1]. The increasing usage of data on cellular networks, driven by the popularity of data-intensive multimedia applications, highlights the importance of these applications for our everyday activities. As the usage of these applications continues to grow, mobile network operators (MNOs) are faced with an increased need to provide their users with a high quality of experience (QoE). Even with the increasing bandwidth of cellular networks with the advent of 5G, MNOs are still unable to meet user expectations for on-demand access to any content at any time and place [2]. This is particularly the case for video streaming. The bandwidth requirement alone can be a challenge for cellular networks during high-quality video streaming. In addition to bandwidth usage, the delivery of multimedia services is sensitive to delay.

Poor QoE can lead to customer dissatisfaction, complaints, and churn. This especially applies to video stream stalling, which is considered the most frustrating form of QoE degradation [3]. Academics, network equipment manufacturers, and MNOs are trying to find solutions to improve network performance and customer QoE. In addition to typical solutions, such as building more dense and efficient mobile networks, researchers are working on improving the visibility of the customer experience.

Therefore, current and future mobile networks should be able to recognize the services used by customers and predict current and future demands for such service delivery. On the MNO side, the recognition of services and QoE guarantees are problematic due to heterogeneous data sources, varying time granularity of collected data, massive data amounts, and stochastic user behavior. Additionally, the increasing use of end-to-end encryption [4] for Internet services makes it more difficult for MNOs to identify which applications their customers are using.

Considering the difficulties of video stream delivery for mobile networks and the limitations of recognizing such a service from the MNO perspective, we propose a method of recognition of mobile video traffic based only on downlink and uplink bitstream. This study continues our previous work [5] on the ability of the convolutional neural network (CNN) to perform classification of data streams on mobile networks generated by various applications running on customers’ devices. We consider the following contributions to the field of traffic classification and service recognition, which can be summarized as follows:We propose a new method for recognizing mobile video traffic based solely on the shape of bitstreams, rather than more complex methods involving the inspection of TCP/IP packets and session flows.We train the CNN using our own collected dataset of real mobile network users.We evaluate our proposed method and demonstrate that it achieves comparable performance to other methods, even when using simpler datasets.

### 1.1. HTTP Adaptive Streaming over Mobile Networks

In mobile 4G and 5G networks, a wireless channel is the last mile of data, video streaming, and voice service delivery. In the wireless channel, the achievable throughput and delay fluctuate rapidly due to (a) the changing quality of the radio frequency (RF) signal, (b) the usage of the shared medium by other clients, and (c) the unknown delay and bandwidth demand of the current user. An unstable wireless environment makes it difficult to guarantee the Quality of Service (QoS) requirements for the services used. HTTP Adaptive Streaming (HAS) was developed in order to mitigate rapid bandwidth fluctuations in the stochastic RF environment.

HAS adjusts video quality parameters to match the current network conditions [3]. HAS necessitates that the video is obtainable in different bitrate versions, that is, in multiple quality levels, and divided into small chunks, each containing several seconds of playing duration. The video stream receiver (also known as the client) monitors the available network bandwidth, buffer status, or both. The next portion of the video is requested at an optimal bit rate to prevent stalling (i.e., the discontinuation of playback due to insufficient data in buffers). In addition to stalling prevention, HAS maximizes the use of available network bandwidth. The advantage of HAS being able to adjust the video to match the available bandwidth has made it a popular choice among providers of over-the-top (OTT) video services such as YouTube, Netflix, Twitch, etc. [6].

In 4G LTE and 5G New Radio (NR) networks, the individual user’s bandwidth depends on the current cell load or the physical resource blocks (PRBs) available in the time-frequency grid [7] and the quality of the RF signal. To maintain stable video playback, users should always have a higher bandwidth than at least the average required by the video codec for a specific resolution plus transmission-related overhead. Every video chunk should be completely downloaded before the previous chunk’s playback time expires; otherwise, stalling occurs.

Such a situation is shown in Figure 1. It represents the downlink bitrate aggregated in 1-s intervals. From the beginning to 70 s, the LTE network’s bandwidth was sufficient for downloading the video chunks at regular intervals. Starting at the 70th second, the download rate became too low for the periodic delivery of video chunks. This resulted in stalling of the video. These measurements were made during video streaming, on the mobile network base station side. The user equipment (UE) was moving toward the poorer Signal-to-Interference and Noise Ratio (SINR) region of the LTE cell. As a result of poorer SINR, modulation and coding efficiency decreased, and transferring the same volume of data required more spectrum resources. Eventually, all available PRB resources were used up by the video stream. After a few seconds of limited bandwidth caused by PRB consumption and low SINR, the playback buffer was depleted and stalling occurred.

The HAS has a periodic requirement to download video chunks, followed by idle periods during which the video chunks are played. If the base stations were aware of the amount of data in each chunk and the download period, the base station scheduler could periodically reserve the required time-frequency resources for this particular user. Therefore, recognition of the video service from the LTE/NR base station side could help to prevent video stalling and avoid QoE degradation for the user.

### 1.2. Encrypted Traffic Recognition

End-to-end (E2E) data transmission is an interaction between the application on the UE side and the server or Content Delivery Network (CDN) on the other side. Typically, CDNs are owned by third-party OTT service providers and not by MNOs. Data delivery is handled using IP stack protocols (HTTP, TPC/IP, UDP, etc.). From a mobile network point of view, it is treated as payload data and is delivered with the best effort, independently of the service used by the client. However, MNOs could benefit from knowing what type of services their clients use. The MNO could then prioritize, block, throttle, or apply different pricing models for specific traffic classes. Techniques such as port-based or deep packet inspection (DPI) were previously commonly employed to identify the type of service or application. However, their accuracy has decreased due to the growing proportion of encrypted traffic [8]. Since 2015, encrypted page loads have increased from under 40% to 83% [9]. End-to-end encryption is a major challenge for recognizing used applications and evaluating the QoE of MNO subscribers. To overcome the challenges of encryption, the literature suggests using indirect methods. Classic machine learning applications for encrypted traffic classification were surveyed in [10], and deep learning methods were extensively surveyed in [11].

### 1.3. Deep Learning for Time Series Classification

Previous studies related to our work propose using TCP/IP packet header or packet interarrival time information to recognize services and applications [12,13,14]. However, this method adds additional complexity related to data preprocessing and requires knowledge of TCP/IP stack protocols. Instead, we propose a simpler approach that uses only the number of bytes transferred and received per second. The bitstream can be considered to be a time series. Such an approach simplifies the task to a time series classification. The categorization of data that is in the form of a time series has been extensively documented in various publications, such as [15,16,17]. In practical applications, a metric as simple as the occupancy of the downlink/uplink buffer is always readily available for the MNO and does not require more complex processing of TCP/IP or QUIC packets or computation of packet-level metrics. Furthermore, by only measuring the number of bytes transferred, customer privacy is not violated, as no content parsing is necessary. Several authors have recommended using deep learning, among other machine learning methods, for identifying encrypted network traffic [18,19]. The authors of [20] proposed using CNN for QoE inference when using various applications on the Internet. Based on the review of related literature, we chose to use CNN in our study.

The remainder of the paper is divided into four sections. The collected dataset and the CNN model used for classification are described in Section 2. Evaluation metrics and testing results are listed in Section 3. In Section 4, we continue the discussion of the obtained results and possible application scenarios. Lastly, we report our conclusions in Section 5.

## 2. Proposed Method

In this section, we describe the method for differentiating video streams from other applications used on mobile networks based on bitstream shape only. We train a CNN for this task on the dataset collected by the authors. The dataset is composed of bitstreams from popular Android applications including Facebook, the browser Chrome, YouTube, and others.

### 2.1. Collected Data Description

The dataset assembled for this study is composed of 3257 h of 1 s granular time series data from real users of the mobile network. Acquiring data from actual users, rather than using synthetic bot-generated data, provides a more precise representation compared with other research [14,21]. The raw dataset includes the following features: the name of the application displayed at the top of the UE display, the number of bytes that were transferred both for download and upload by the network module, and a timestamp. The Android application is used to collect the name of the onscreen application and the network buffer information from a few volunteers. The model is trained using information collected from the UE side. The same data are always available at the base station of the mobile network and can be used for inference. Using only the number of bytes downloaded and uploaded makes it possible for mobile network operators (MNOs) to use this model without directly violating privacy or incurring the high computational costs of parsing encrypted TCP/IP packets.

### 2.2. Dataset Generation

The next step is to construct a dataset to train the neural network. To preserve valuable information, we divide the collected data into windows of 60, 300, and 600 s, with overlaps of 59, 299, and 599 s, respectively. We assign a label to every data window indicating the name of the application running on the smartphone at that time. We discard windows that contain multiple applications.

The windowing data augmentation technique produced a massive quantity of training samples, including over 9 million samples that are 60 s long, over 5 million samples that are 300 s long, and over 3 million samples that are 600 s long. From this amount of data, we can select a balanced number of samples representing each application.

### 2.3. Dataset Categories Description

To gain insight into the usage patterns of the collected data, we analyze it in terms of duration of use and statistical properties. We present the applications with the longest usage duration in Table 1. Over 28% of the samples are generated during inactive phone usage and have the Start Screen label. Even when the smartphone is not in use, applications continue to generate background data transfer. Facebook is the next most frequently used application. The third and fourth ranked are Chrome and Google Internet browsers. The list continues with YouTube, a video streaming platform, followed by 9GAG, which is an image-downloading application.

We select and visualize a few random examples of the most used applications. In Figure 2, we visualize five-minute-duration data samples that represent downloading and uploading bitstreams when YouTube, Facebook, Chrome, Outlook, Messenger, or 9GAG apps are in use.

By analyzing random samples, we can infer that different applications have unique bitstream forms. For example, the Chrome, Facebook, and 9GAG apps show periods of high data transfer and intervals of minimal network activity. We can infer that these patterns correspond to periods when the user is actively engaging with the app and downloading content, and when they are viewing or reading previously downloaded content. The HAS pattern is clearly observable in the YouTube application, as video segments are both downloaded and played. The emailing application Outlook uses both downloading and uploading simultaneously, to receive and send emails. The last example is the Messenger app. It displays a pattern of constant download and upload during voice or video calls.

Random examples do not represent all of the cases and are used to visually understand possible time series shapes generated by different applications. To make comparisons across a large number of data samples, we need to use statistical features of the data. For this reason, we use the library of time series feature generation [22] to generate statistical features to compare YouTube samples (which we select as the representation of the HAS video service) against all other applications. For our purposes, we consider only the mean and standard deviation. Figure 3 shows the kernel density estimate (KDE) of the mean and standard deviation calculated from the generated samples. We use logarithmic representations of the *x* axis for a more accurate visualization.

We can see that there is a partial overlap between the means and standard deviations of the samples. On average, video has a higher mean bitrate and a higher standard deviation within the sample. Additionally, YouTube has multiple peaks, which are probably due to video streams of different resolutions.

An additional interesting point is the fundamental frequency of video bitstream samples. Figure 4 shows the cumulative distribution function of the downlink bitrate during adaptive video streaming. When we look at the bitstream shape of the video stream as a signal, we can observe a fundamental frequency of around 10 s. This fundamental frequency shows the delivery period of the video chunks or the duration of the playtime of each chunk. After recognizing that video streaming is being used, the mobile network base station could learn other parameters such as the average bit rate, video chunk size, and fundamental frequency. This would allow the base station of the mobile network to configure the scheduler for video service delivery with a QoS guarantee.

Although the selected statistical features (mean and standard deviation) show distinctions between video and other services, further differences can be drawn by considering time series shapes.

### 2.4. Convolutional Neural Network for Video Stream Recognition

Convolutional networks are neural networks that employ convolution in at least one of their layers, rather than using general matrix multiplication [23]. CNNs are optimized to handle data in array form [24]. Our dataset is a collection of 1D (if only downlink is considered) or 2D (downlink and uplink) arrays. For this reason, CNN is used to recognize the video stream only by looking at the received and transmitted bytes. The diagram of the proposed neural network is shown in Figure 5. We utilize the Keras library to implement the design and training of the CNN model [25]. We choose to use the deep learning architecture to evaluate the feasibility of video recognition only, without conducting a thorough examination of the deep learning algorithm or optimizing its parameters.

The CNN uses a vector of one or two rows as its input, which represents downloaded-only or downloaded and uploaded bytes, and consists of 60, 300, or 600 numerical values. The input vector undergoes convolution, followed by batch normalization [26]. Nonlinear Rectified Linear Unit (ReLU) functions are then applied to the normalized data. The process of convolution, normalization, and ReLU is repeated three times using multiple layers. After the last ReLU function, Global Average Pooling is applied. Finally, the last layer produces the distribution of probability between video streams or other applications. The deep learning model is trained using Adam Optimizer and the sparse categorical cross-entropy loss function. Training progress is evaluated using a sparse categorical accuracy. After training, the model is evaluated against new data.

The preparation of the dataset incurs a high memory requirement, and the model training demands GPU resources. We use the Google Cloud Platform to deliver high-performance and flexible computing resources.

## 3. Results

In this section, we describe our testing methods, evaluation metrics, and the results of testing for video stream recognition from the bitstream shape only.

The video stream recognition model is trained by separating samples of 60, 300, and 600 s into two categories: video and the rest. The video category is composed only of samples from the YouTube app, even though the HAS service can be used inside the Internet browser or other applications. For any sample sequence length, we select 100,000 samples to represent video and 100,000 samples from all other applications. Selecting an equal number of samples from each category helps avoid training with imbalanced data. The model is trained and evaluated using different sequence lengths, looking only at downlink bytes, next downlink, and uplink in 1D sequence (uplink sequence added at the end of downlink) and 2D sequence (uplink and downlink fed simultaneously).

### 3.1. Evaluation Metrics

To evaluate the performance of the classification, we use several evaluation metrics. First, we calculate the accuracy using Equation (Equation 1). Then, we determine the precision and recall using Equation (Equation 2).
(1)$$\begin{array}{c} Accuracy=\frac{T_{p}+T_{n}}{T_{p}+T_{n}+F_{p}+F_{n}} \end{array}$$
(2)$$\begin{array}{cc} P=\frac{T_{p}}{T_{p}+F_{p}},\; & R=\frac{T_{p}}{T_{p}+F_{n}} \end{array}$$

Accuracy is calculated as the number of correct classification predictions divided by the total number of predictions. Precision *P* is defined as the number of true positives $T_{p}$ divided by the number of true positives plus the number of false positives $F_{p}$. Recall *R* is defined as the number of true positives $T_{p}$ divided by the number of true positives plus the number of false negatives $F_{n}$.

Furthermore, we calculate the misclassification rate for each of the most used applications according to Equation (Equation 3). The misclassification rate shows the number of instances that are incorrectly classified out of the total instances. In our case, it represents the number of instances where the YouTube application, which is classified as a video streaming service, was misclassified as another application, and vice versa for the other most used applications (Facebook, Chrome browser, etc.).
(3)$$\begin{array}{c} Misclassification\;rate=\frac{F_{n}+F_{p}}{T_{n}+T_{p}+F_{n}+F_{p}} \end{array}$$

Using the misclassification rate for each of the most used applications, we verify not only how many times the video is correctly or incorrectly recognized, but also which applications confuse our model.

### 3.2. Performance Evaluation

The model is evaluated using input sequences with identical duration to those of the training data, but with completely new and unseen samples. Samples are chosen at random from the gathered dataset in equal proportions from the video and the rest of the applications. The classification accuracy achieved by the trained CNN model in classifying 60, 300, and 600 s sequences is shown in Table 2.

The precision and recall in distinguishing between the video stream and other applications, using the CNN to classify 60, 300, and 600 s sequences, are shown in Table 3.

To add more robustness to the evaluation and increase proximity to the real mobile network usage scenario, we evaluate our model against the new data in the same proportions as collected. The number of samples from each application and their misclassification rates are presented in Figure 6.

### 3.3. Comparison with Other Methods

In this section, we present the results of a comparison of the proposed method with other techniques that aim to identify the class of services or specific applications that transfer data over the network. The comparison is based on various criteria, such as the purpose of classification or recognition, the algorithm type, the method of dataset collection or generation, and the complexity of data preprocessing and required networking domain knowledge. Finally, the performance of the methods is compared in terms of accuracy, precision, and recall, and the results are presented in Table 4.

It is worth noting that other related works use datasets that contain more information, including data from inside Internet Protocols such as source port, destination port, number of bytes in a packet payload, TCP window size, interarrival time, and packet direction. In contrast, our proposed method only evaluates the shape of bitstreams generated by different services on a second-level granularity, without inspecting the content of packets. If the aim is not only to recognize encrypted video streaming but also to guarantee QoS/QoE from the mobile network base station side, then data availability must be considered. The scheduler of LTE and NR nodes is implemented in lower layers, such as MAC, which are not aware of TCP/IP packet content and treat it as payload only. Therefore, we believe that proposing a method of identifying video streaming services based solely on the shape of the generated bitstream would enable us to use simple and accessible data without raising privacy concerns.

The proposed method simplifies network traffic classification by eliminating the need for TCP/IP packet parsing and session flow construction. Furthermore, the proposed method does not necessitate knowledge of the TCP/IP protocol stack and its implementation in actual network nodes. The comparison shows that despite the use of simple bitstream input data, the proposed method is still comparable with the other methods in terms of accuracy, precision, and recall.

## 4. Discussion

Based on the evaluation results presented in Table 2, we observe that the precision of recognizing the video stream from other applications is relatively low for an input sequence of 60 s, and reaches 90% for input sequences of 300 s and longer. We also see that using downlink and uplink data gives better performance than relying only on downlink bitstreams. In LTE and NR networks, the base station controls downlink and uplink scheduling; including uplink bitstream does not represent a limitation to the practical implementation of the proposed method. The precision and recall results in Table 3 also suggest that a duration of 60 s is too short for recognizing that a video streaming service is in use. As shown in Figure 4, the fundamental frequency of HAS is around 10 s, so only a few chunks are downloaded and acknowledged during the one-minute interval.

Based on the misclassification rate results of the 300 s input sequences shown in Figure 6, we can see that YouTube is the top application in terms of usage time in our test dataset, and it is correctly recognized in 85% of predictions. Misclassifications mainly occur during the times when HAS was not streamed, but the application start page was browsed. Other applications with high misclassification rates are 9GAG and Facebook. Both applications, in addition to picture browsing, have embedded video players. In our work, misclassification between video and non-video services mostly occurs due to the capabilities of the applications, such as when the video application is only being browsed, or when another application is using a video streaming service.

The results obtained show that distinguishing between video streaming and other types of applications is feasible, but the dataset collection should be improved to avoid times when the application with the video streaming capability is in use, but no video is being streamed.

## 5. Conclusions

With the increasing use of encrypted traffic and VPNs, it is becoming difficult to determine the applications and services that customers use on mobile networks. However, recognizing the service being used is crucial to ensure specific QoE requirements. Furthermore, identifying and categorizing mobile data traffic based on service or application can help prioritize specific types of traffic, implement dynamic pricing, or block MNOs.

In this study, we investigate the feasibility of identifying video streaming services solely based on the shape of the generated bitstream. MNOs can collect bitstream data at all times without violating privacy concerns. Furthermore, the use of a convolutional neural network (CNN) model does not require data preprocessing or extensive knowledge of data transfer protocols.

The findings of this study indicate that the CNN has the ability to distinguish between video streams and other types of data streams produced by various applications. The high demand for video streaming suggests a requirement for future work that aims not only to recognize encrypted video streaming, but also to guarantee QoS/QoE. Knowing that a client is engaged in video streaming and being aware of the parameters of the HAS service can aid in optimizing resource scheduling at the mobile network base station.

## Figures and Tables

**Figure 1 sensors-23-02548-f001:**
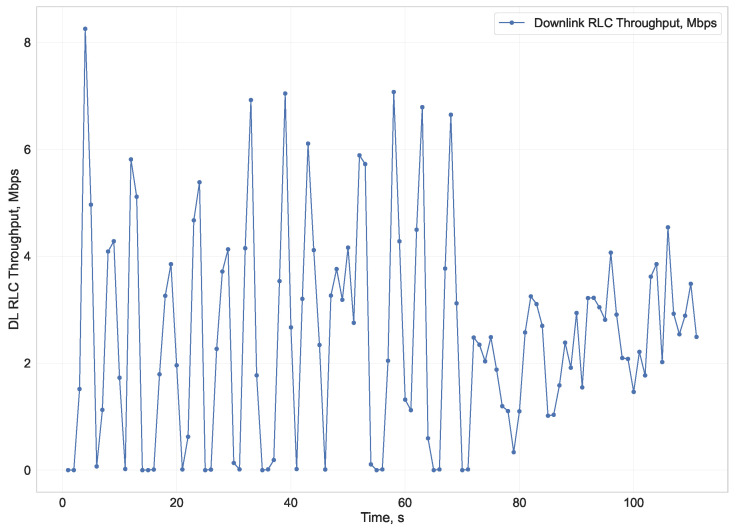
Measured downlink bitrate from mobile base station side during video stream with stalling.

**Figure 2 sensors-23-02548-f002:**
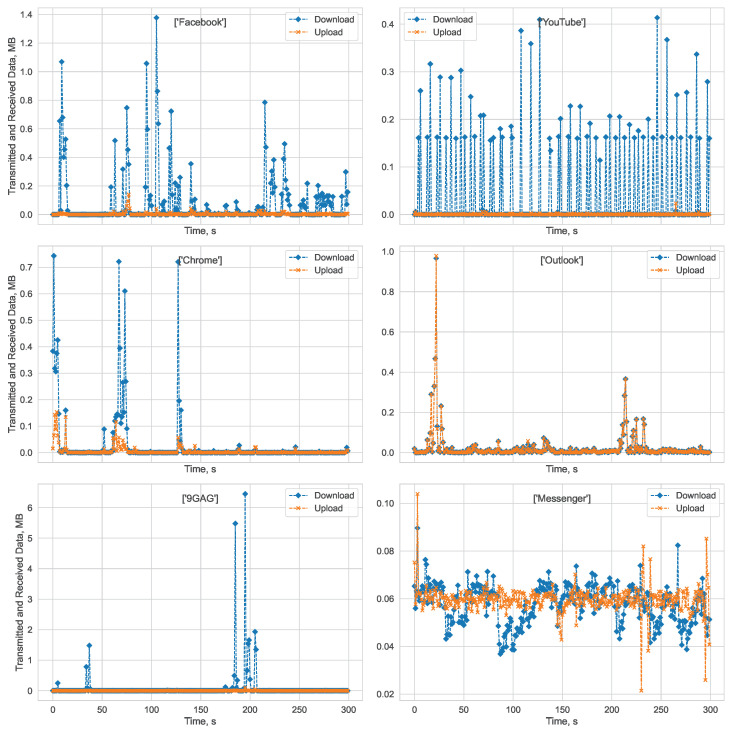
Random samples of bitstreams representing different applications.

**Figure 3 sensors-23-02548-f003:**
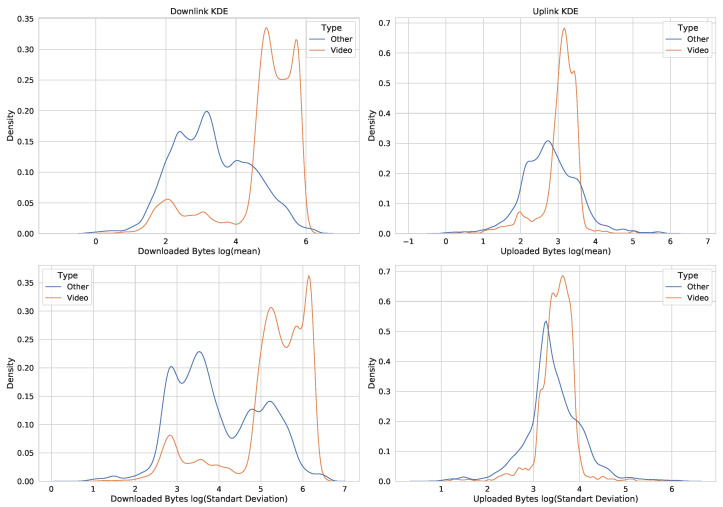
Statistical evaluation of the collected dataset: KDE of the means and standard deviations calculated from the generated samples, separated by the type of service used.

**Figure 4 sensors-23-02548-f004:**
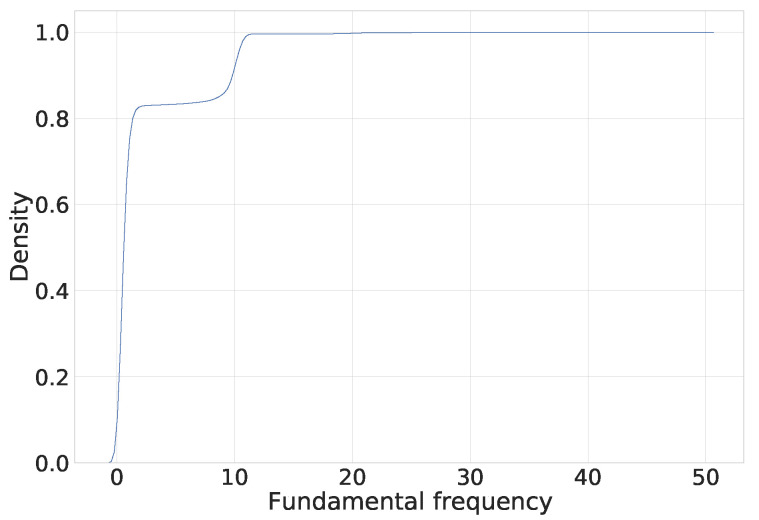
The fundamental frequency cumulative distribution function of the downlink bitrate during adaptive video streaming.

**Figure 5 sensors-23-02548-f005:**
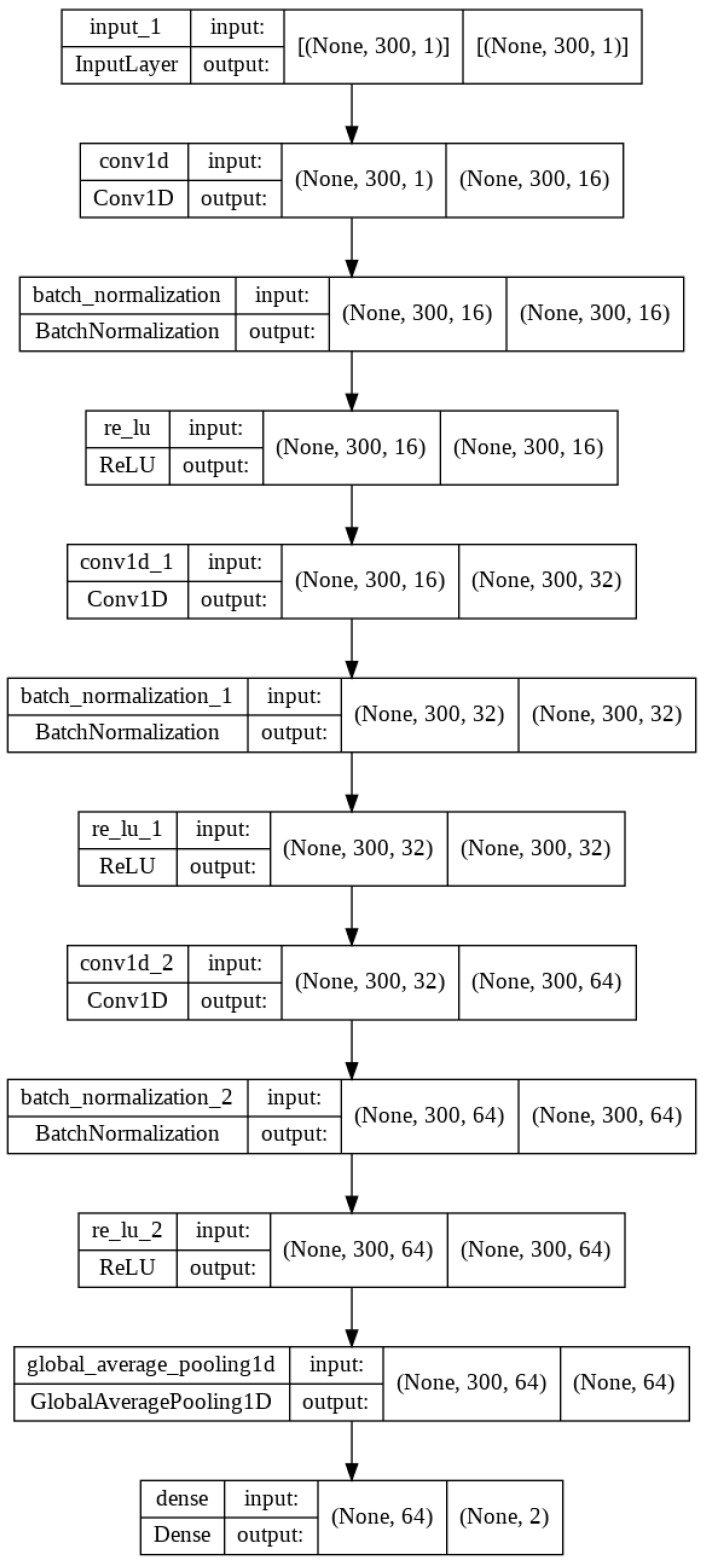
Diagram of the convolutional neural network employed for video traffic recognition.

**Figure 6 sensors-23-02548-f006:**
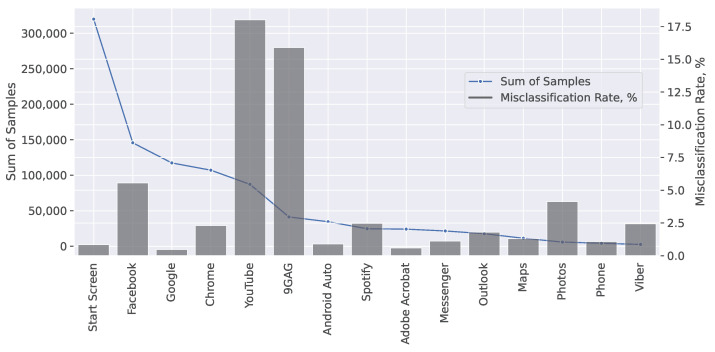
Misclassification of the Video category as other applications, and other applications as the Video category, by specific application.

**Table 1 sensors-23-02548-t001:** Applications ranked by usage duration.

*Application on Screen*	*Duration Percentage, %*	*60 s Sample Count*
Start Screen	28.63%	2,672,802
Facebook	13.87%	1,294,841
Chrome	11.26%	1,051,053
Google	9.62%	898,295
YouTube	6.59%	614,714
9GAG	3.15%	293,852
Android Auto	2.77%	258,592
Messenger	2.55%	238,133
Outlook	2.30%	214,883

**Table 2 sensors-23-02548-t002:** Test Accuracy.

Test Accuracy			
Sequence Length	Downlink Only	Downlink and Uplink 1D	Downlink and Uplink 2D
60 s	0.772	0.814	0.837
300 s	0.859	0.902	0.896
600 s	0.815	0.9	0.929

**Table 3 sensors-23-02548-t003:** Precision and Recall.

		Precision			Recall		
		60 s	300 s	600 s	60 s	300 s	600 s
Downlink Only	Video	0.902	0.991	0.986	0.726	0.799	0.860
	Other App	0.769	0.830	0.875	0.921	0.993	0.988
Downlink and Uplink 1D	Video	0.974	0.969	0.983	0.679	0.816	0.985
	Other App	0.752	0.840	0.879	0.982	0.974	0.866
Downlink and Uplink 2D	Video	0.965	0.982	0.997	0.711	0.806	0.900
	Other App	0.770	0.834	0.908	0.974	0.985	0.998

**Table 4 sensors-23-02548-t004:** Comparison with other methods.

Work	Year	Algorithm	Dataset	Complexity (Inputs and Preprocessing)	Accuracy	P	R
[27]	2022	Transformer	ISCX VPNnonVPN	High. Packet (pcap) format files used. TCP/IP knowledge is required.	0.993	0.988	0.988
[28]	2019	LSTM	ISCX VPNnonVPN	High. Network packet flow. TCP/IP knowledge is required.	0.912	-	-
[19]	2020	Autoencoder	ISCX VPNnonVPN	High. Header and payload. TCP/IP knowledge is required.	0.98	0.99	0.99
[13]	2017	CNN	RedIRIS	High. IP flow headers. TCP/IP knowledge is required.	0.961	0.952	0.961
Proposed method	2023	CNN	Collected by Authors	Low. Bitstream shape. TCP/IP knowledge not required.	0.929	0.997	0.900

## Data Availability

Our dataset was generated during the study. Initially, the collected data are available here: https://drive.google.com/file/d/19VIeauKmQoyBVlJi2sP5C0ma0ml7TuXx/view?usp=sharing accessed on 22 February 2023.

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
