# Peer review of "Video Stream Recognition Using Bitstream Shape for Mobile Network QoE"

_sensors, 2023, doi:10.3390/s23052548_

Round 1

Reviewer 1 Report

The study proposes a method to recognize video streams transferred in a cellular network based on the shape of the bitstreams. The manuscript is of good quality, it is well written and has a good list of references.

There is however a major drawback. Since this topic is well studied, there are many existing network traffic classification methods available, and therefore some comparison is expected to be conducted when a "new" method is introduced (if we can call new using CNN for network traffic classification). In this study, such a comparison is missing. I would recommend either compare the method with existing methods that also only use bytes downloaded/uploaded if there are any, or compare with the approaches that use more features to show that despite using only bitrate your approach is  still comparable with those in terms of accuracy, precision, recall, etc. Until such a comparison is carried out, the manuscript's contribution to the field of study is close to zero.

Other minor issues:

Figure 1 is of low quality, vector graphics formats are recommended to be used. 

Tense is not consistent throughout the manuscript, e.g. Section 2 starts with past tense than switches to present, then goes back to past, I am not native speaker, but I guess it is not grammatically correct :) and it just sounds weird. Present is usually used in articles. 

Subsection 2.2 - how many 600 seconds long samples?

Subsection 2.3 - if the browser is used, can the user use it to open a video streaming website, e.g. YouTube? How is the traffic categorized in this case?

Figure 2: y ticks of some subplots overlap, please fix.

Figure 3, is subplot title missing for the left column? Should it be downlink KDE? 

Page 6: "datashown"

Figure 4's font size is obviously different from the rest, it is barely visible.

Figure 6 is of low quality, vector graphics should be used.

Author Response

Dear Reviewer,
Thank you for taking the time to review our work. We appreciate your feedback. Your comments and suggestion also will be considered in our future works. The reply to the highlighted issues is attached.

Thank you.

Reviewer 2 Report

In this manuscript, I give my assessment of Research Paper entitled “Video Stream Recognition Using Bit-Stream Shape for Mobile Network QoE". The paper presents original research results & have acceptable originality with thorough study. The proposed technique seems correct, as validated by simulation graphs, however there are some suggestion to improve the paper

1) Add a remark in the introductory section which summarize the novelty of proposed work.

2) The flow chart in Fig. 5 need to explained further.

Author Response

Dear Reviewer,
Thank you for taking the time to review our work. We appreciate your feedback. The reply to the highlighted issues is attached.

Thank you.

Reviewer 3 Report

The authors are investigating the possibility to use CNN to differentiate between video traffic flow and other flows in a mobile network environment. As the authors state, the issue of flow detection is crucial in the QoS/QoE domain and it becomes increasingly difficult due to the rising share of encrypted traffic flows. The authors formed their database containing various bitstream flows and employed CNN for categorization.

I suggest enriching the paper by using your solution to other datasets to show how you compare with the methods of other authors, or vice-versa, use the methods of other authors on your dataset. This would yield a better understanding of how you compare with others. Otherwise, it is difficult to evaluate how successful your solution is.

Introduce a legend in Figure 6.

Author Response

(The authors gave the same response as above.)

Round 2

Reviewer 1 Report

Thank you for taking my comments into account, the manuscript looks much better now.

It is however worth noticing that the comparison with other methods is expected to be conducted using the same dataset. Why cannot you just implement the methods used for comparison and test them on your dataset to calculate accuracy and other metrics is a mystery to me :) Here it is probably fine, but not a good practice in general, as such comparison does not make much sense.

Author Response

Thank you once again for your time and effort in reviewing our manuscript. The answer to the raised issue is attached. Your comments additionally will be considered in future research designs.

Reviewer 3 Report

In the previous review round, I suggested using other methods on the author's dataset or using the author's method on other datasets for comparison purposes. The authors only provided the evaluation results of other studies that used different datasets, so it is still difficult to rate the accuracy of the author's approach.

Author Response

(The authors gave the same response as above.)
